# Strengths and Weaknesses in the Risk Management of Blood-Borne Infections: Qualitative Research in Public Health

**DOI:** 10.3390/ijerph17186650

**Published:** 2020-09-12

**Authors:** Anita Gębska Kuczerowska, Artur Błoński, Joanna Kuczerowska, Robert Gajda

**Affiliations:** 1National Institute of Public Health,24 Chocimska str., 00-791 Warsaw, Poland; arturblonski@wp.pl; 2Collegium Medicum, Cardinal Stefan Wyszynski University, Kazimierza Wóycickiego 1, 01-938 Warsaw, Poland; 3Department Internal Medicine, Endocrinology and Diabetology, Bródnowski Hospital, Medical University in Warsaw, Kondratowicza 8, 03-242 Warsaw, Poland; jkuczerowska@gmail.com; 4Gajda-Med Medical Center in Pułtusk, ul. Piotra Skargi 23/29, 06-100 Pułtusk, Poland; gajda@gajdamed.pl

**Keywords:** blood-borne infections, risk management, strengths, weaknesses, public health, Data-Aided Process Enhancement and Repair

## Abstract

This article presents the research from the first phase of our study on blood-borne risk management, wherein we solicited epidemiologists’ and healthcare practitioners’ expert opinions on a blood-borne infection risk assessment in Poland. Forty-two experts were recommended by epidemiology consultants and recruited from all districts in Poland. We used the SWOT (Strengths, Weaknesses, Opportunities, Threats) method in the evaluation. Experts’ opinions showed that there is room for improvement in the prevention of blood-borne infections. Commonly reported weaknesses in the risk assessment included gaps in knowledge and inappropriate procedures, which are largely caused by financial constraints and practitioners’ lack of awareness of developments in their trade. Strengths included legal regulations for medical services and procedures, surveillance, and increasing awareness on the part of medical staff. When paired with the existing statistical data, these results provide a comprehensive view of the problem of blood-borne infections in Poland. The analysis supported the development of a strategy proposal to prevent blood-borne infections and enhance existing risk assessment procedures.

## 1. Introduction

Statistical data on blood-borne infections are processed and used within the sentinel system. Databases on health indicators, covered by the obligation to report all diagnosed cases, are a crucial element of this surveillance system. The statistical data help measure the endpoints (e.g., of the population health status as the effects of multiple processes); however, the data provide no direct information on the processes. The methods of population health status measurements have their methodological limitations, as well as the limitations for statistical data. At the same time, knowledge of the problems and discrepancies (on processes and health indicators) at different levels and kinds of public health services allows for continual improvement, as per the well-known rule “hominis est errare, insipientis in errore perseverare” [1]. This is particularly important for public health because of the unmeasured, potential risk of blood-borne infections in nonmedical services, where procedures are conducted with less epidemiological supervision than is required in the healthcare service sector.

Despite progress in the diagnosis and treatment of blood-borne infections, data from the World Health Organization (WHO) demonstrate that the problem is still significant [2]. Our familiarity with the mechanisms of blood-borne infections—the well-known pathogens, sources of infections, and pathways through which infections can spread [3,4,5] and the established microbiological etiology of blood-borne infections (HBV-Hepatitis B Virus, HCV-Hepatitis C Virus, HDV-Hepatitis D Virus, and HIV-Human Immunodeficiency Virus) [6]—sometimes leads us to underestimate the problem [7]. According to research on viral pathogen analyses, the risk of blood-borne infections (expressed as a pooled transmission rate) was largest for HBV (1.45%), followed by HCV (0.46%) and HIV (0.0056%) [8,9]. When considering the viral etiology of blood-borne infections, the global epidemiology data indicated that, in 2015, 1.75 million people were infected with HCV, in 2017, 1.1 million people were newly infected with HBV [10], and in 2016, 162,471 people died from viral hepatitis and its complications [11]. The problem of blood-borne infections (including viral infections) is a burden for the public health, and we must analyze the costs of viral hepatitis and other etiologies and their impacts on the health system [2,12], especially in those countries where blood-borne infection prevention presents a particular challenge [13,14]. The complexity of blood-borne infection risks necessitates preventative policy solutions, permanent surveillance in the service sector, and research on the unknown risk factors.

The results summarized here derive from the first phase of a research project funded by the Swiss Contribution and the Polish Ministry of Health (MoH) [15], which was completed in July and August 2016. This phase of the project focused on analyses of current public health practices (by statistical data and other evidence) for the prevention of blood-borne infections, intending to find solutions to identify in detail problems in management that were partly confident. Opinions were collected from experts in the field of epidemiology and practitioners providing medical and nonmedical services (including those of hair and beauty salons, wellness centers, and tattoo salons) whose daily work involved a risk of blood-borne infections. Since we sought to analyze systemic needs in public health services, we selected experts who could perform both epidemiological and organizational analyses and who could collectively address the problem of infection across the country from a practical point of view. The analysis covered many aspects, including the state of knowledge, the way existing regulations were implemented, how incidents of exposure to blood-borne infections were monitored and reported, and the relevant financial and educational issues. Experts also commented on how standards developed and implemented in the medical service sector could be transferred to the nonmedical service sector. In this way, we were able to assess the key challenges of blood-borne infection prevention in the service sector.

This qualitative research project allowed us to gather information that might be useful for Poland’s strategy proposal program for the prevention of blood-borne infections presented in a detailed report, such as selected expert opinions on the risk assessment and current blood-borne disease prevention in Poland.

## 2. Materials and Methods

Since we sought to analyze the problem of blood-borne infection risk assessment from a management point of view, we decided to apply a qualitative research approach [16,17,18]. However, another research team was invited to perform a parallel epidemiology research and analyses of statistical data concerning blood-borne infections in Poland (in selected risk groups and representative studies on the prevalence of HCV). Our study was preceded by a pilot interview with regional epidemiology consultants, wherein we considered what data to gather for the scope of the research. An external company commissioned professional gathering interviews. We informed the facilitator of the research issues before the study, and we ensured that the respondents did not recognize this facilitator. During the interviews, the research team observed the discussions to extrapolate a model (with the possibility to moderate additional new aspects of this issue).

Since we sought to analyze the problem of blood-borne infection risk assessment from the perspective of different experts and practitioners scattered across all regions of Poland, we carried out the project online [19,20] using the research platform “Idea Blog”. This method allowed us to engage a large number of professionals within a specific but flexible time frame, eliciting diverse opinions that could enrich our data. Further, it allowed the participating experts to choose the place and time of their consultation; this flexibility, combined with the guarantee of anonymity, enabled us to collect more responses than would otherwise have been possible.

Forty-two experts and practitioners, representing state institutions and the private service sector, completed the survey. Experts represented all 16 regions of the country (with a similar number of experts per voivodeship). The vast majority of respondents were with higher/and additional postgraduate education (worked professionally as: management staff, practitioners, epidemiologists, and other service providers).

The eligibility criteria for the experts’ elections were:at least one expert per region,at least 5 years of professional experience,a regional epidemiological consultant or an individual recommended by such a consultant (not random selection),professional activities in the field of epidemiological surveillance (e.g., sanitary inspection), andprofessional activities in the medical and nonmedical service industries and representatives of NGOs (Non-Government Organization*s*) (dealing with issues relating to the high risk of blood-borne infections).

Experts were invited to participate in the study by mail, which was sent by the Public and Epidemiology Country Consultants from the National Institute of Public Health. This correspondence included information on the aims and scope of the research and, also, the consent. Participation was voluntary, and informed consent was obtained before the study. The research obtained ethical approval from a steering committee and abided by the rules of the Declaration of Helsinki (Project KIK35). As per the eligibility criteria, the respondents represented all 16 provinces in Poland. All of their statements were anonymous and were assigned a three-letter, randomly selected code. The statements were presented in the original wording, and only the identifying information (including regional data) was removed. In the period from 18 July to 20 August 2016, the forum participants were asked to give their opinions daily. Through 3 to 4 days, the data were collected in individual interviews (the theme in Table 1); in the last 1 to 2 days, the interview data were opened to discuss on IdeaBlog by all. For each task, experts illustrated their answers with facts, statistical and bibliographic data, concrete examples, and solutions. The source data, statistics, and research evidence confirming all expertise were simultaneously verified by the research team. The question examples and themes are presented in Table 1.

The opinions collected were analyzed using the SWOT method to ascertain the strengths, weaknesses, opportunities, and threats associated with the risk management of blood-borne infections. The themes included legislation, funding, and the organization of the services system. The analysis also covered (1) aspects of education; (2) the form/content of information delivered to the public, service recipients, and service providers, respectively; and (3) practice examples, particularly those that are recommended for optimal prevention. In this paper, we focus on the responses that clarify the strengths and weaknesses of blood-borne infection risk management in public health (due to the large scope of the study). In the separate one concerns the issues of opportunities and threats in blood borne infections risk management.

## 3. Results

In this first phase of our study, we discussed with experts the strengths and weaknesses of the public health management of blood-borne infections in the medical and nonmedical service sectors. The results are presented in Table 2.

These results provided a basis for the next phases of our study (II—assessing the discrepancies in management and III—discussing the results with policymakers). The study’s aim for the first and second phases was collecting specialists’ opinions on infection risks and its management. The third phase was to inform policymakers on the study’s results and prepare proposals for the prevention of blood-borne infections in Poland.

## 4. Discussion

The authors presented the use of the qualitative method in conjunction with process-oriented data mining and a statistical analysis of the management process for blood-borne disease risks in medical and nonmedical services. During the discussions, our experts presented their knowledge and experience using examples; they proposed their thoughts and suggestions for solutions. A broad review of the data, combined with management interpretation, provided the opportunity to exchange knowledge between experts on the IdeaBlog platform (debate). At the same time, the developed material was a rich source of additional supporting information. Guaranteeing confidentiality as to the content of the supplementary information and the anonymity of the respondents allowed us to get to know the problem in detail to build the most accurate solution proposals. The information presented is only a summary of the selected and most important issues, without details in evidence. The results of this study point to some imperfections in the current system for the prevention of blood-borne infections. Broadly, the problems arise due to cost savings, where costs associated with prevention are minimized in response to economic factors but without considering the potential costs of treatment and compensation that may arise in future cases of infection of the staff or patients [2,21,22]. The latter also extends to accessing restrictions that make it difficult to directly consult specialists in the case of exposure to pathogenic agents, even though such exposure requires rapid preventive action [23,24]. On the one hand, this limitation is systemic—based on the economic decisions by the fund limitations at the disposal of the public payer—but at the same time, it points to the locally organizational nature. If infection prevention becomes a high priority for managers, it will be easier to allocate available resources for that purpose. Additionally, it seems that the time of the pandemic COVID-19 is forcing everyone to respect a sanitary regime, although this statement would yet require scientific proof.

Related to economic factors are also functional limitations of the infection prevention system, as respondents pointed out the lack of practical training in this field. Even with the appropriate theoretical knowledge, only repetitive actions can become habitual and effective, mainly since many countries have observed insufficient compliance with post-exposure recommendations. Medical staff are often not interested in updating their theoretical knowledge in the field of infection prevention, because this does not translate directly into new healthcare options, preferring to train in new clinical skills, which further limits the effectiveness of the system. Other researchers have emphasized that the lack of knowledge amongst general practitioners may be especially problematic, as they cover the largest group of patients; as such, this is the group where the educational potential is the highest [24].

The greatest weakness of the current model is medical staffs’ ongoing defensive attitudes towards patients with potential blood-borne infections, who are consequently treated as the possible source of future infections [25]. The latter limits actions to individual protection and, to a lesser extent, patient protection, which may increase the risk of infection (for example, when a practitioner uses one pair of disposable gloves for several patients). Our respondents and other researchers drew attention to staffs’ limited awareness of the risk, particularly those operating in healthcare [26]. Meanwhile, in the cosmetic, hairdressing, tattoo, and piercing sectors, the risk of infection may be higher than in medical facilities due to the lack of mandatory training for employees [27,28,29]. A crucial observation of the epidemiology experts and practitioners in our study showed that they largely agreed to point at the strength of the law system—of the basic rules for medical services together with the EU regulations. According to respondents, all these acts create a coherent law system of guidelines and do not require any additional actions.

The strength of the current system is the organizational and legal strengths of infectious disease hospital teams in many institutions that provide medical services. Overall, these teams operate at high levels, lead the education, supervise, monitor staff, and thereby, actively shape the prevention and prophylaxis of infections. Another positive phenomenon is the increasing knowledge of the medical staff. According to the respondents, this is the result of improved education and vocational training, both in-person and online, that is financed by hospitals and foreign (primarily the EU and Swiss Contribution) bodies. Furthermore, new forms of education—such as improved access to education programs on the internet—allow practitioners to pursue self-training and independently search for educational materials. Participants also credited the practical elements of training tailored to specific services and occupational groups. These solutions are not systematic, being mostly local and ad hoc in nature; still, they allow for the identification and correction of errors in the daily operations of public health.

A crucial observation shared by the majority of the participants in the study was the stable epidemiological situation of blood-borne infections like hepatitis C. While new HCV infections continue to occur, there is no steep growth—that is, it is “under control”—which proves that the prevention system works in most cases but still requires sealing.

Our study fills a gap in the existing knowledge about the functioning of medical and nonmedical service sectors by clarifying experts’ perspectives on the health security and risk management for blood-borne infections. The study supplemented a statistical analysis of risk assessment based on epidemiological data, as well as the case studies. The qualitative research methods seem to be necessary to provide context for the findings of process-oriented data mining. Routine analyses of epidemiological data is part of the sentinel system, but each method has its limitations, such as the lack of information (regarding causes or otherwise) on record for unregistered/undiagnosed cases. From the perspective of public health, it is important to know these limitations and to update the risk information continually. Alongside the existing statistical data, the evidence from this study provides a comprehensive view of the problem of blood-borne infections in Poland from the perspective of epidemiology and public health. Statistics provide a basis for making rational decisions, but the qualitative analysis provides grounds for strongly supporting these decisions. A report from this study was issued to decision-makers at various levels of management in the medical services sector. The information has enhanced education programs and vocational training for medical and nonmedical professionals.

## 5. Conclusions

The study emphasized the results of combining different research approaches (quality and supported by data from quantity) by using process-oriented data mining (e.g., infection risk and its prevention). The analysis shows that there are still serious gaps relating to the management of blood-borne infections in the medical and nonmedical service sectors. In this regard, there were no regional differences. These gaps were evident both in the epidemiological data and in the statements provided by experts and service providers, who either confessed their ignorance and other restrictions (e.g., financial) or talked about known bad practices in their industry/service sector. The epidemic situation is stable thanks to improved screening and restriction rules for new cases; the number of infections is growing, but all experts agree that this is attributable to better diagnosis rather than ongoing mismanagement. Fortunately, most blood-borne infections are curable, but not all of them are preventable by immunization. It is assumed that not all pathogens are well-diagnosed or known. As such, there is always a reason for better prevention and education programs.

## Figures and Tables

**Table 1 ijerph-17-06650-t001:** Themes and examples questions (the same for all participants).

**(1) The role of blood-borne infections in the participants’ daily work** Describe the range of occupational activities and the relevance of the problem of blood-borne infections in the workplace. How time-consuming are the issues of blood-borne infections in the workplace? Information on a traineeship on the topic of infection and, specifically, the infectious disease area (useful for the planning of education program content). Assess the overall epidemiological situation in the surveyed area.
**(2) Awareness of the risk of blood-borne infections** Assess public awareness of blood-borne infections amongst: practitioners (doctors, nurses, beauticians, and hairdressers); patients/clients (e.g., of beauty salons and tattoo salons); and decision-makers. Assess the importance/significance of the problem with justification (the provision of examples/facts). What attitudes do people express towards infected people and the problem of blood-borne infections?
**(3) Rules to reduce the risk of blood-borne infections** Assess the current legislation in the area of (blood-borne) infections. Specify the legal provisions and justify positive and negative assessments. Which areas of the law need to be improved to ensure sanitation and safety? What good practices would you recommend from your own experiences and the experiences of others?
**(4) Rules and life: How are the guidelines implemented in practice?** Evaluate service providers’ practices regarding the legal framework. What are the most common violations of the law on sanitary safety for each of the following: medical services and beauty services (hairdressers, beauticians, and tattooists). Evaluate the results of audits/inspections designed to ensure adherence to the applicable regulations (and any relevant statistical data). What are the offenses/crimes committed by institutions and companies? What are the barriers that may explain the failure of these legal provisions? What solutions might you propose to improve the applicable law (for example, amendments and additional records)? Evaluate the awareness of patients/clients and service providers about the risk of blood-borne infections and describe their attitudes towards recommended prevention procedures. Assess where there may be opportunities to improve practitioners’ (and service providers’) compliance with recommendations and regulations.
**(5) Supervision and incident reporting of staff and patient/client infections** Provide some context about rating scales, incidents of injuries, and the risk of infections in your workplace. Comparatively, assess local problems in the region and country. Comparatively, assess local problems against those of other countries. What are the causes/reasons for occupational exposure, and how does the management play the role of risk (e.g., prevention/post-exposure procedures)? Evaluate: the reasons for recording exposures to infectious agents, the reasons for reporting and not disclosing exposures, and the factors conducive to good record keeping. Evaluate the lack of preventive and corrective actions (and the role of the employer or client/patient in this regard). Assess the technical and logistical procedures for reporting infections. What suggestions would you make to improve the effectiveness of the surveillance system?
**(6) Good practices observed in daily work** Provide information on local infection prevention programs. Which activities would you assess positively and recommend? Provide information on possible sources of funding for activities related to the prophylaxis/prevention of HBV and HCV.
**(7) Education and training about blood-borne infections** Provide information on local educational actions, including: training on blood-borne infection prevention; implementation time and executor of the educational program; the recipients of the educational program and their degree of interest; feedback on the educational program; and feedback on the learning outcomes (knowledge, attitudes, and behavior). Provide information and opinions on the manner and forms of education related to blood-borne infections.
**(8) The role of finances in the prevention of blood-borne infections** Evaluate the budget and the degree of funding available for the problem of infection. Evaluate the financial limits for the scope of the proper prophylaxis/prevention of infections. Evaluate the deficits and financial savings in the medical and nonmedical sectors, as well as their implications (risks).
**(9) The importance of the disinfection process for reducing blood-borne infections** Describe the theory and practice of disinfection as expressed/performed by medical and nonmedical personnel whose work presents a risk of infection. What are the criteria for the selection, purchase, and use of disinfection measures and equipment? What are the implications of these criteria for savings?
**(10) The role of patients/clients in prophylaxis/preventing blood-borne infections** What positive infection-prevention behaviors have patients/clients exhibited that are worth promoting and strengthening? Provide some context about the patients/clients in question (justification, scope, and media). Evaluate the services market, including the pricing for medical and nonmedical services.

**Table 2 ijerph-17-06650-t002:** SWOT (Strengths, Weaknesses, Opportunities, Threats) (first phase analysis): Assessment of weak and strong aspects of blood-borne infection prevention in public health services.

Tasks Framework	Strengths	Weaknesses
Policy level	Medical services law regulations are well-designed. Coherence of the law system and guidelines. The stable epidemic situation (before pandemic). Effective control of the blood-borne infections in the healthcare system. *** Increasing the offer for education programs and training for medical staff in the area of infection risk. Free access to knowledge through education programs. E-learning and web access to knowledge saving time and associated costs. Practical forms of local training adapted to the specific nature of services and professional groups. ***	Management’s attitude limited to fulfilling only legal obligations (negligence or misuse of safety procedures in hospital management). ** In the general population, the needs are limited to updates and broadening the knowledge in health protection. There is no obligatory training for some service staff in nonmedical services. The other service providers were not obliged to develop knowledge in this issue as medical staff.
Local organization and management level	Highly qualified epidemiology staff in most hospitals dedicated to controlling the infections. Effective control of the blood-borne infections in the healthcare system. *** Practical forms of local training adapted to the specific nature of services and professional groups. ***	Priority of the economic reasons to choose in management: cheaper suppliers to increase personal savings, cheaper and less effective cleaning products, and out-sourcing—using the services (like cleaning companies) without adequate training. Knowledge gap between medical staff in the hospital and PHC (Primary HealthCare). For staff of nonmedical services, the continuing education was assessed as the loss of profit and time. The quality and safety of services have less value on management than procedure. The stigmatization of infected patients (longer waiting time).** Difficulty accessing medical care (diagnosis and treatment) because of contagious patients. ** Insufficient attention of the staff to hygiene(before pandemic). Management’s attitude limited to fulfilling only legal obligations (negligence or misuse of safety procedures in hospital management). **
Medical staff level—Individual and Team	Mostly local and ad hoc in nature of the actions taken for the identification and correction of errors. Routine actions taken (education, supervising, and monitoring) by epidemiology teams in hospitals. Practical forms of local training adapted to the specific nature of services and professional groups. *** Effective control of the blood-borne infections in the healthcare system. ***	The attitude of medical staff to prioritize the prevention of the infection risk to ourselves. The stigmatization of infected patients (longer waiting time). ** Lack of practical knowledge among staff on the transmission of infection. Difficulty accessing medical care (diagnosis and treatment) because of contagious patients. ** The gap between knowledge and practical actions—attitude and behaviors on the infection risk. Many practitioners underestimated the problem of blood-borne infections, citing it as a “problem for doctors” (long practice without needle sticks).

** Tasks for 2 levels. *** Tasks for all levels.

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
