# Peer review of "Strengths and Weaknesses in the Risk Management of Blood-Borne Infections: Qualitative Research in Public Health"

_ijerph, 2020, doi:10.3390/ijerph17186650_

Round 1
Reviewer 1 Report
Attached are editorial changes in a "track changes" format. Please navigate to the "Review" tab, and select "All Markup." Navigate through the comments using "Previous" and "Next."

Author Response
Dear Reviewer:
We thank you for your thoughtful suggestions and corrections. The manuscript has benefited from these insightful suggestions.
The manuscript has been rechecked and the necessary changes have been made in accordance with the recommendations. Thank you very much to the Reviewer for your valuable help in improving the article by the editorial changes.
Kind regards
Authors
Reviewer 2 Report
- At first of the introduction, some blood-borne infections should be mentioned as examples.
- If the authors analyzed the opportunities and threats, they should describe the results of OT.
- Did the authors ask all questions in Table 2 to all participants? They should describe the methods about asking questions in more detail.
Author Response
Dear Reviewer:
We thank you for your thoughtful suggestions and insights. The manuscript has benefited from these insightful suggestions.
The manuscript has been rechecked and the necessary changes have been made in accordance with the suggestions. The responses to all comments have been prepared and given below.
- 1. At first of the introduction, some blood-borne infections should be mentioned as examples.
Response:
1. We thank the Reviewer for this comment. We have revised the manuscript to include in the introduction as follows:
“Our familiarity with the mechanisms of blood-borne infections—the well-known pathogens, sources of infections, and pathways through which infections can spread [3–5]; the established microbiological etiology of blood-borne infections (HBV, HCV, HDV, and HIV) [6]—sometimes lead us to underestimate the problem [7].”
(page 1-2, lines 44-48)
- If the authors analyzed the opportunities and threats, they should describe the results of OT.
Response:
We thank the Reviewer for this comment. We have included additional information about our methodological approach to present the results.
We have revised the manuscript as follows:
“In this paper, we focus on the responses that clarify the strengths and weaknesses of blood-borne infection risk management in public health (due to the large scope of the study). In the separate one concerns the issues of opportunities and threats in blood-borne infections risk management.”
(page 5, line 131-134)
- Did the authors ask all questions in Table 2 to all participants? They should describe the methods of asking questions in more detail.
Response: We thank the Reviewer for this suggestion.
As requested, we have additionally included as follows in the section - Materials and Methods:
(page 3, lines 118-124)
In the header of the table, there is additional information in brackets
- “Table 1. Themes and examples questions (the same for all participants).”
(page 3, lines 125)
We thank you for your suggestions. The manuscript has benefited from these insightful corrections.
Kind regards
Authors

Reviewer 3 Report
This is a well written article describing a study that investigates the risk management of blood borne infections in Poland. The analysis is described appropriately and the findings may be useful in educating health care professionals in strategies for infection risk assessment and prevention.
Author Response
Dear Reviewer:
We thank you for your excellent opinion of our article.
This is a well-written article describing a study that investigates the risk management of blood-borne infections in Poland. The analysis is described appropriately and the findings may be useful in educating health care professionals in strategies for infection risk assessment and prevention.
Response:
The results of the study were used to prepare (rationally) educational programs and to propose strategies in the risk management of blood-borne infections in Poland. The project has been implemented with great success. Thank you very much to the Reviewer for appreciating our efforts and work.
Kind regards
Authors
Reviewer 4 Report
In this paper the authors reported a qualitative analysis on blood-borne risk management collecting the opinions of forty-two experts about risk and prevention of blood-borne infections. These opinions are analyzed using the SWOT method and the ensuing results were reported and discussed in this paper.
This is an original paper with some interesting results and in my opinion this manuscript would be suitable for publication previuos some revisions:
- Firstly, in “Materials and Methods” section it is reported that “Forty-two experts and practitioners, representing state institutions and the private service sector, completed the survey.”; then the eligibility criteria are reported but it is not clear the differentiation of the sample of experts according to the kind of occupation and regional origin. It could be an interesting information to be included in the paper.
- In Table 2 is reported in some aspects the term “before pandemic”. I suggest to add in the “Discussion” section some comments about this, in particular if the above-mentioned aspects are still applicable even in the pandemic era.
- Table 2 is structured as a list of strengths and weaknesses. For a better understanding of this qualitative analysis I would suggest to group the highlighted aspects according to a thematic subdivision (see for example: Vincent C, Understanding and Responding to Adverse Events, Health Policy Report. The New England Journal of Medicine, 2003, 348;11).
- In “Conclusion” section the authors reported that “the study emphasized the results of combining different research approaches (quality and quantity)”. I think that the quantity approach in not mentioned enough in the text. Although it is not the focus of this study, maybe it would be better to include some information about this aspect.
- Finally, there are some English and typing error that need to be corrected.
Author Response
Dear Reviewer:
We thank you for your thoughtful suggestions. The manuscript has benefited from these insightful suggestions and corrections.
The manuscript has been rechecked and the necessary changes have been made in accordance with the recommendations. The responses to all comments have been prepared and given below.
- Firstly, in the “Materials and Methods” section it is reported that “Forty-two experts and practitioners, representing state institutions and the private service sector, completed the survey.”;
- then the eligibility criteria are reported but it is not clear the differentiation of the sample of experts according to the kind of occupation and regional origin. It could be an interesting information to be included in the paper.
Response:
1.We thank the Reviewer for this comment. We have revised the manuscript to include clarification regarding the methodology - additional information on experts - as follows:
“Forty-two experts and practitioners, representing state institutions and the private service sector, completed the survey. Experts represented all 16 regions of the country (with a similar number of experts per voivodeship). The vast majority of respondents were with higher/and additional postgraduate education (worked professionally as: management staff, practitioners, epidemiologists, and other service providers).”
(page 3, lines 97-101)
- In Table 2 is reported in some aspects the term “before pandemic”. I suggest to add in the “Discussion” section some comments about this, in particular if the above-mentioned aspects are still applicable even in the pandemic era.
Response:
We thank the Reviewer for this comment. We have revised the manuscript in the “Discussion” part by adding the suggested information as follows:
“On the one hand, this limitation is systemic— based on the economic decisions by the funds' limitations at the disposal of the public payer—but at the same time, it points to the locally organizational nature. If infection prevention becomes a high priority for managers, it will be easier to allocate available resources for that purpose. Also, it seems that the time of the pandemic COVID -19 – forces everyone to respect the sanitary regime, although this statement would require scientific prove yet.”
(page 9, line 165-170)
- Table 2 is structured as a list of strengths and weaknesses. For a better understanding of this qualitative analysis I would suggest to group the highlighted aspects according to a thematic subdivision (see for example: Vincent C, Understanding and Responding to Adverse Events, Health Policy Report. The New England Journal of Medicine, 2003, 348;11).
Response:
We thank the Reviewer for this suggestion.
As suggested, we have grouped the results in table 2. According to thematic subdivisions based on task framework (policy level, local management level, and staff management level). The proposal for the new table - as follows:
(page 6-8)
TASKS FRAMEWORK |
|
STRENGTHS |
WEAKNESSES |
|
|
|
|
Policy level |
|
· Medical services law regulations (Act of 2008, EU) are well designed · Coherence of the law system and guidelines. · The stable epidemic situation. (before pandemic) · Effective control of the blood-borne infections in the healthcare system.*** · Increasing the offer for Education programs and training for medical staff in the area of infection risk. · Free access to knowledge through education programs · E-learning and web access to knowledge save time and associated costs · Practical forms of local training adapted to the specific nature of services and professional groups.***
|
· Management's attitude limited to fulfilling only legal obligations (negligence or misuse of safety procedures in hospital management) ** · In the general population, the needs are limited to update and broaden knowledge in health protection. · There is no obligatory training for some service staff in non-medical service. · The other service providers were not obliged to develop knowledge in this issue, as medical staff.
|
Local organization and management level |
|
· Highly qualified epidemiology staff in most hospitals dedicated to controlling the infections · Effective control of the blood-borne infections in the healthcare system.*** · Practical forms of local training adapted to the specific nature of services and professional groups.***
|
· Priority the economic reasons to choose in management : - cheaper suppliers to increase personal savings, - cheaper and less effective cleaning products - out-sourcing - using the services (like cleaning companies) without adequate training. · Knowledge gap between medical staff in the hospital and PHC · For the staff of non-medical services, the continuing education was assessed as the loss of profit and time · Quality and safety of services have less value on management than the procedure. · The stigmatization of infected patients (longer waiting time)** · Difficulty accessing medical care (diagnosis, treatment) because of contagious patients. ** · Insufficient attention of the staff to hygiene(before pandemic) · Management's attitude limited to fulfilling only legal obligations (negligence or misuse of safety procedures in hospital management) ** |
Medical staff level -Individual and Team |
|
· Mostly local and ad hoc in nature of action taken for identification and correction of errors. · Routine actions taken (education, supervising, monitoring) by the epidemiology team in hospitals · Practical forms of local training adapted to the specific nature of services and professional groups.*** · Effective control of the blood-borne infections in the healthcare system.***
|
· The attitude of medical staff to prioritize the prevention of the infection risk to ourselves · The stigmatization of infected patients (longer waiting time)** · Lack of practical knowledge among staff on the transmission of infection. · Difficulty accessing medical care (diagnosis, treatment) because of contagious patients.** · The gap between knowledge and practical actions - attitude and behaviors on infection risk · Many practitioners underestimated the problem of blood-borne infections, citing it as a “problem for doctors”. (Long practice without needle-sticks)
|
** tasks for 2 levels
*** tasks for all levels
- In the “Conclusion” section the authors reported that “the study emphasized the results of combining different research approaches (quality and quantity)”. I think that the quantity approach in not mentioned enough in the text. Although it is not the focus of this study, maybe it would be better to include some information about this aspect.
Response:
We thank the Reviewer for this suggestion.
As requested, we have included additional information in methods and in conclusions, as follows:
“In the period from 18 July to 20 August 2016, the forum participants were asked to give their opinions daily. Thru 3-4 days the data were collected in individual interviews ( the theme in tab.1) in the last 1-2 days the interview data was opened to discuss on IdeaBlog by all. For each task, experts illustrated their answers with facts, statistical and bibliographic data, concrete examples, and solutions. The source data, statistics, and research evidence confirming all expertise were simultaneously verified by the research team.”
(page 3 lines 118-123)
“The study emphasized the results of combining different research approaches (quality and supported by data from quantity studies) by using process-oriented data mining (e.g., infection risk and its prevention). The analysis shows that there are still serious gaps relating to the management of blood-borne infections in the medical and non-medical service sectors. In this regard, there were no regional differences. (..)”
(page 10 lines 226-230)
- Finally, there are some English and typing error that need to be corrected.
Response:
We thank the Reviewer for this suggestion.
In accordance with the suggestions, English and typing error have been corrected.
We thank you for your all suggestions. The manuscript has benefited from these insightful corrections
Kind regards
Authors

Round 2
Reviewer 2 Report
The manuscript has been revised according to our comments.